# Next Step in Gene Delivery: Modern Approaches and Further Perspectives of AAV Tropism Modification

**DOI:** 10.3390/pharmaceutics13050750

**Published:** 2021-05-19

**Authors:** Maxim A. Korneyenkov, Andrey A. Zamyatnin

**Affiliations:** 1Faculty of Bioengineering and Bioinformatics, Lomonosov Moscow State University, 119991 Moscow, Russia; maxkorneenkov@mail.ru; 2Institute of Molecular Medicine, Sechenov First Moscow State Medical University, 119991 Moscow, Russia; 3Department of Biotechnology, Sirius University of Science and Technology, 1 Olympic Ave, 354340 Sochi, Russia; 4Belozersky Institute of Physico-Chemical Biology, Lomonosov Moscow State University, 119992 Moscow, Russia

**Keywords:** AAV engineering, AAV tropism modification, gene delivery, gene therapy

## Abstract

Today, adeno-associated virus (AAV) is an extremely popular choice for gene therapy delivery. The safety profile and simplicity of the genome organization are the decisive advantages which allow us to claim that AAV is currently among the most promising vectors. Several drugs based on AAV have been approved in the USA and Europe, but AAV serotypes’ unspecific tissue tropism is still a serious limitation. In recent decades, several techniques have been developed to overcome this barrier, such as the rational design, directed evolution and chemical conjugation of targeting molecules with a capsid. Today, all of the abovementioned approaches confer the possibility to produce AAV capsids with tailored tropism, but recent data indicate that a better understanding of AAV biology and the growth of structural data may theoretically constitute a rational approach to most effectively produce highly selective and targeted AAV capsids. However, while we are still far from this goal, other approaches are still in play, despite their drawbacks and limitations.

## 1. Introduction

Gene therapy (GT) is a relatively novel and promising branch of biomedicine which is based on the delivery of a transgene encoding either therapeutic protein or RNA. The application of GT-based drugs may be extremely useful in many different branches of therapy, such as cancer therapy, and the treatment of infectious diseases, monogenetic diseases, and many others.

Among the most important issues of GT is the target delivery to cells of interest. Today, several systems of delivery are utilized, such as viral vectors, nanoparticles and liposomes. Viral vectors are considered as one of the most promising gene delivery systems due to the wide diversity of cell attachment and entry mechanisms.

Different viral vectors have different cell tropisms, genome capacities and safety profiles; hence, different viruses could potentially be used for targeted delivery. Modern research in this area is focused mostly on retroviruses, adenoviruses and adeno-associated viruses [1]. Adenoviruses have a huge insertion capacity, which varies from 1–2 to 35 kb for gutless vectors; they do not integrate into the genome, but, on the other hand, adenoviruses have a low safety profile due to their high immunogenicity to lethal cases caused by systemic administration and the short transgene expression time [2]. 

Retroviruses were the first to be applied for gene therapy. Moloney murine leukemia virus has been utilized to transduce tumor-infiltrating lymphocytes (TILs) with a neomycin phosphotransferase gene using the LNL6 vector [3]. Retroviruses possess a short duration of transgene expression and low safety due to cases of genome integration that lead to induced oncogenesis [4]. Lentiviruses induce insertional mutagenesis less frequently than retroviruses but do not exclude it completely [5].

In comparison to the viral vectors listed above, adeno-associated viruses (AAVs) have a better safety profile. This is crucial for the successful clinical application of gene therapy. At least two Food and Drug Administration (FDA)-approved drugs based on AAV are currently manufactured: Luxturna, which is used for the treatment of Leber congenital amaurosis, and Zolgensma for spinal muscular atrophy [6]. AAV genome capacity is 4.8 kilobases, which is lower in comparison to other vectors; it is unable to replicate viral DNA on its own, so it is preferable to use AAV for the targeting of slowly dividing cells, for example, fibroblast-like synoviocytes (FLSs) and macrophage-like synoviocytes (MLSs), which are considered to be promising targets for rheumatoid arthritis gene therapy [7]. AAV tissue tropism varies depending on the type of strain, as different strains may have different receptors. Meanwhile, broad tissue tropism represents a serious limitation towards target gene delivery and simple GT usage. Hence, the alteration and modification of tropism could help to overcome these tropism limitations.

This review summarizes different approaches of AAV retargeting and tropism modifications, their limitations, accomplishments and several difficulties that may be faced by anyone who decides to alter the AAV structure to generate a selective GT.

## 2. AAV Biology

Adeno-associated virus belongs to the *Parvoviridae* family, which consists of two subfamilies: *Parvovirinae,* which includes viruses that infect vertebrate, and *Densovirinae,* which infect invertebrates [8]. *Parvoviridae* members are non-enveloped viruses with a single-stranded 4–6 kbp DNA genome comprising two open reading frames (ORFs): *rep*, which encodes viral replicative proteins, and *cap*, which encodes three viral proteins, VP1, VP2 and VP3 (Table 1). The plus and minus strands are both found in separate viral capsids. The genome is flanked by two 145 bp inverted terminal repeats (ITRs). ITRs form T-like hairpins with 125 bp and 20 bp form D-sequences that is important to the high efficacy of AAV DNA rescue and replication processes [9]. 

ITRs have a rep-protein binding site (RBS) and a terminal resolution site (TRS), which works as the origin of replication. ITRs are also required for integration into the host genome at chromosome 19 in humans [9,45] and are crucial for successful AAV single-stranded DNA (ssDNA) encapsidation [46,47,48]. The first ORF contains promoters P5 and P19 and encodes proteins Rep78, Rep68, Rep52 and Rep40 (Figure 1). Rep78 and Rep68 expression is regulated by the P5 promoter. These proteins are required for DNA replication, site-specific integration, the regulation of gene expression and the excision of AAV DNA from the host genome [49]. Both Rep78 and Rep68 can bind to the ITR hairpin and cleave at the TRS. Rep78 can also initiate host cell arrest in the S phase and facilitate latent infection [50]. Rep68 is an endonuclease which takes part in DNA replication initiation, site-specific integration and transcription regulation [51,52]. The P19 promoter facilitates Rep52 and Rep40 expression. These proteins possess 3′ to 5′ helicase activity and are essential for viral genome packaging [53,54].

The right ORF encodes three capsid proteins: VP1, VP2 and VP3, with a molecular weight of 87, 72 and 62 kDa, respectively. The P40 promoter initiates the transcription of the Cap gene and produces two mRNAs by alternative splicing [55]. The unspliced transcript encodes VP1; VP2 is encoded by spliced mRNA with an alternative start codon ACG [56]. VP3 is encoded by spliced transcript with a conventional ATG codon and, furthermore, is essential for correct capsid assembly [57,58]. Sixty copies of VP1-VP3 form an icosahedral capsid with a size of approximately 23–28 nm in a molar ratio of 1:1:10 [59]. The N-terminus of VP1 also possesses phospholipase A2 (PLA2) activity which is required for AAV virion escape from endosomes, and contains a nuclear localization signal (NLS) [60]. The Cap gene also contains a frameshifted ORF that encodes assembly-activating protein (AAP). This protein is required for VP protein transport to the nucleolar region and capsid assembly [61].

To date, 12 strains of AAV have been characterized (Table 1) based on a phylogenetic analysis [62,63]. These serotypes show diverse tropism characteristics due to a variety of cellular primary and co-receptors. The crystal structures of eight AAV strains have been characterized [64,65]. Pseudotyping experiments indicate that capsid proteins are exclusively responsible for tissue tropism [66,67]. Different serotypes of AAV have different origins of isolation. Thus, AAV serotypes 1, 3, 4, 7, 8, 10 and 11 were isolated from non-human primates, and serotypes 2, 5, 6 and 9 were isolated from humans (Table 1). Strains attach to the cell surface via interactions with primary and co-receptors. For example, heparan sulfate proteoglycan (HSPG) is the primary receptor for AAV2 [68], but some human-derived AAV2 strains do not use HSPG as the receptor and lack arginine residues at the position 585 and 588 [69]. AAV structure studies have progressed since pioneering works on this topic, and much more is currently known about AAV attachment [70,71,72]. Nonetheless, some serotypes require more study, as they may possess some useful characteristics and novel tropism variants.

## 3. Capsid Engineering for Tropism Modification

AAV capsids have good properties for gene delivery applications. Although AAV serotypes cover many different tissue tropism variants, sometimes, it is not sufficient for the successful development of clinically approved therapy, because most of the abovementioned serotypes lack tissue specificity, and some tissues are not within AAV’s tropism spectra. To overcome these tropism limitations, several approaches have been applied to date. These approaches may be artificially separated into three groups: rational design, directed evolution and chemical conjugation.

### 3.1. Rational Design

The rational approach was the first to be applied for AAV capsid modification. This method utilizes the direct alteration of capsid structure via peptide or protein insertions (Figure 2). As the insertion of a novel sequence may alter the capsid protein structure, causing the inability to assemble capsids, it is crucial to search for insertion sites in non-conserved putative loops that should be exposed on the surface of a capsid and should be located in hydrophilic regions. These assumptions allowed several sites to be discovered that have been successfully utilized for AAV retargeting.

One of the first attempts utilized a combination of plasmids encoding wild-type VPs and VPs with the incorporation of a 29.4 kDa single-chain fragment variable region of antibody (sFv) against CD34 into the N-terminus of VPs. The VP2-sFv-containing strain showed increased transduction of CD34+ myoleukemia cell line KG-1 but was unable to assemble independently. This issue was resolved by the combined use of sFv-VP-2 protein and VP1-3 wild-type protein expression, but viral titer was relatively low (4 × 10^2^ infective virions/mL), and the composition of viral capsid was more likely to differ from the standard ratio of 1:1:10 of VP1, VP2 and VP3, respectively [73]. The insertion of a serpin receptor ligand KFNKPFVFLI into the VP2 N-terminal site (aa138) and FVFLI into VP1 (aa34) showed a 15- and 64-fold increase in infectivity on IB3 cells, and the capsid assembly process did not require wild-type VP proteins, thus showing that relatively small insertions are well tolerated [74]. Those located in flexible regions allowed several sites to be discovered which are suitable for the insertion of L14 motif QAGTALRGDNPQG, which recognizes the integrin receptor, but only one of these six sites, I-587 (numbering from AAV2 VP1 start codon), showed increased transduction of AAV2-resistant cell cultures Co-115 and B16F10 [75]. The same position was also used for the insertion of SIGYPLP, which targets endothelium cells. The expression of the reporter gene in some of the endothelium cells was ~6- and ~28-fold higher in comparison to the wild type and was also independent of heparin binding [76]. In addition to previously mentioned research, other studies have been conducted using this site, rendering it a common choice for AAV rational engineering [77,78]. There are also some insertion sites in putative loops III and IV in positions 449 and 588 that were proved to exhibit the increased transduction of CD13+ cells, such as RD and KS1767. Both the replacement of the wild-type sequence with targeting peptide NGR and the insertion of a targeting peptide in the position 588 site showed a 10- to 20-fold increase in the transduction of RD and KS1767 cells. Targeting peptides were identified using the phage display method, which showed that the peptides with double cysteine flanking the sequence possess higher binding efficiency [79]. The length of targeting peptide is suitable for successful retargeting. Thus, the insertion of more than 30-40 aa into the I-587 site may lead to an increase in empty capsid production or the misfolding of VP2. This can be solved by the deletion of abundant aa residues from the putative loop, preventing it from elongation and folding alterations. For example, Reid et al. [80] attempted to insert a minimized Z34C protein A binding domain at the 587 site with and without deletion. A mutant with an insertion and deletion of 9 aa showed almost similar levels of packaging efficiency compared to the wild type, while the mutant with insertion only produced more empty capsids. Both mutants bound immunoglobulins successfully, but the mutant without deletion bound more effectively.

In some cases, there are no data concerning retargeting peptide sequences due to the lack of information concerning specific cell receptors. This problem can be resolved by the usage of peptide libraries and subsequent selection of peptides with the best properties. Muller et al. [81] utilized a multiplasmid system that expresses wild-type VP proteins and VP proteins with insertion at position 3967 with the purpose of creating a vector able to transduce human coronary artery cells. The plasmid pMT187-0-3 was used as a template for insertions and was designed in a way that the plasmid had a shifted ORF so it could not express proteins correctly if it did not contain an insertion. This was achieved by the prior insertion of a restriction site with the extra nucleotide. To prevent the simultaneous expression of more than one peptide insertion, the first generation of capsid was produced in 293T cells with a mix of insertion-containing plasmids and wild-type plasmid without ITRs, so the latter was unable to be packaged into capsid; hence, such a chimeric capsid would contain only plasmid encoding one specific insertion, which would be expressed on the second-generation capsids. To select AAV capsids with a higher tropism, Muller et al. [81] transduced primary coronary artery cells with the AAV library. As not every peptide insertion managed to transfect cell culture, only capsids with the best tropism were able to infect cells. After several rounds of target cell transduction insertion, motifs of enriched AAV capsids were amplified by PCR, and at least three peptide motifs with increased binding to target cells were observed.

Not every AAV retargeting attempt is focused on insertion techniques; some try to alter single amino acid residues. Different strains of AAV demonstrate different tissue tropisms due to non-conserved amino acid residues. Thus, the analysis and substitution of non-conserved residues may lead to enhanced or altered tropism. Bowles et al. [82] engineered a chimeric strain with high muscle tropism using the AAV2 strain as a template, as AAV2-based therapy has been approved for clinical application. Viral protein sequence alignment allowed the identification of AA residues which possess higher muscle tropism that should also be located in variable regions of the capsid surface and differ from those in the AAV2 sequence. Three chimeric strains were generated, but only AAV 2.5 showed good results in the muscle transduction test, as it showed 1.8–5.5-fold higher transduction in comparison to AAV2, while two other strains did not show any significant increase or showed it at day 42 after injection. AAV 2.5 contained four AAV1 substitutions (N705A, Q263A, V708A and T716N, AAV2 numbering) and one insertion of T265 from AAV1. In addition to the high transduction ratio, substitution and insertion did not affect yield production, heparin binding or decreased recognition by A20 antibodies. Moreover, both anti-AAV1 and anti-AAV2 mouse antibodies neutralized the infectivity of AAV2.5 with 5-fold decreased effectiveness. This shows that several AA substitutions may alter the antigenic properties of capsids greatly. 

Boucas et al. [83] attempted to retarget AAV2 using a new insertion site 453 and RGD-4C sequence, which targets αVβ3 integrin. Compared to wild-type AAV2, modified AAV2 showed lower transduction efficiency. RGD-4C insertion in 587 also showed lower targeting molecule binding, but amino acid substitutions R585A and R588A, which are known to abolish HSPG binding, led to 50- and 33-fold increases in binding for 453 and 587 insertion variants, respectively. This shows that sometimes even correct insertion may show unimpressive results that might be corrected with additional mutations. Such data also indicate that interactions between the capsid and target proteins are not completely clear.

Another issue that should be taken into account by everyone who strives for the production of a recombinant vector is post-translational modifications of viral capsid proteins. For example, VP1 and VP3 proteins of 1, 2, 5, 7, 9 and rh10 strains contain N-terminal acetylation. The influence of N-terminal acetylation on viral capsid proteins is still not clear; some suggest that the N-acetylation of VP1 and VP3 may influence viral capsid degradation and uncoating before entry into the nucleus [84]. To elucidate whether N-acetylation affects transduction properties, Frederick et al. [85] yielded AAV5 mutation variants S2G, S2P, S194P, S194G, S2G/S194G and S2P/S194P. Each mutation led to the substitution from serine to either glycine or proline, which are less prone to N-terminal acetylation after the degradation of N-terminal residues by aminopeptidases [84,86], and compared their transduction properties with a wild-type strain in mice retina. As a result, no acetylation was observed in S to P variants, and only 10% acetylation in S to G mutants was observed. These mutants had an assembly efficiency similar to the wild type. Regarding transduction efficiency, all acetylation variants showed reduced transduction levels, but the AAV5S194G variant showed specific and high GFP expression in photoreceptors at a dose of either 1 × 10^8^ or 5 × 10^8^ in comparison to wild-type AAV5; however, there was no statistically significant difference at a dose of 1 × 10^9^, probably due to the saturation of eGFP expression in photoreceptors.

The influence of deamidation was also evaluated. The deamidation of a side-chain amide group is a common protein degradation signal. The AAV2G58D deamidation mutant showed a similar eGFP expression level in comparison to wild-type AAV2 after the transduction of retinal ganglion cells, but AAV2N57D showed a decreased eGFP expression level and a significantly decreased level of vector genome copies. These results show that the influence of PTS on viral capsid properties is possible, but there is still much to understand, as different modification sites sometimes influence capsid properties differently.

### 3.2. Directed Evolution

Directed evolution (DE) is an alternative approach for the production of proteins with tailored properties; hence, it may be applied to produce a viral vector for target gene delivery. The main tools for this approach are techniques that induce mutagenesis in protein encoding genes. Different methods currently exist, for example, site-directed mutagenesis, error-prone PCR, gene shuffling (Figure 3) and a combination of rational design and the evolutional approach. DE is especially useful in cases when the absence of data concerning the physicochemical properties of structures and proteins restricts the application of a pure rational approach. Gene shuffling was the first method to be applied for a directed evolution of AAV capsid. Schaffer and Maheshri [87] applied the staggered extension process (StEP) [88] for the library generation of the AAV2 CAP ORF. This step is based on modified PCR with short cycles and quick elongation of about 100 bp per cycle. Short fragments synthesized with this technique are shuffled with other strands and allow new genes with different sequences to be obtained. The resulting library of genes was used to produce novel AAV capsid variants to evade immune response or alter heparin affinity. Capsid affinity was tested by heparin column chromatography. This test revealed capsids either with increased or decreased tropism, and DNA sequencing showed various novel mutations in the CAP ORF. Wild-type AAV was eluted with 450 nm NaCl solution, while variants with increased tropism were eluted with higher concentrations. The transduction properties of the novel capsids were also analyzed with cell cultures HEK293 and Chinese hamster ovary (CHO) in the presence of a soluble heparin and two cell cultures with a reduced expression of heparan sulfate, where the same results were obtained [87,89].

Since directed evolution techniques are time consuming due to the possibility of enormous numbers of variants, it is still useful to combine it with a structure-guided approach where possible. A competent structural analysis facilitates library generation and narrows its size by screening out the worst and unsuitable variants. For instance, a combination of the structure analysis of AAV9 capsid and error-prone PCR allowed tens of AAV9-derived variants to be obtained with simultaneous altered tropism to several tissues and decreased liver transduction by the mutagenesis of only GH loop aa390–627 (VP1 numbering), as it is surface exposed [90].

With the purpose to develop mouse liver-targeting vectors, Marsic et al. [91] compared the structural and sequence data of 150 AAV2 variants and found highly variable sequences on the surface of a capsid which then were mutated to enrich the AAV library. To avoid incompatible residue substitution, the pool of amino acid residues was restricted to those which appeared at least once in a given position among parental capsids. Additionally, four more positions were rationally modified: Y444F and Y500F to increase transduction efficiency through a directed reduction in the interaction with the proteasome of degradation, and R585 with R588 were modified to eradicate capsid interaction with primary receptor HSPG. Although the capacity of the primary library was evaluated as 1 × 10^8^, only one AAV2-derived variant Li-C containing mutations Gly263Ala, Thr503Pro, Tyr500Phe and Lys507Thr was extracted. Li-C liver transduction efficiency was much higher than AAV2 and comparable to AAV8 levels.

The generation of AAV capsids library is also possible with the use of the SCHEMA algorithm. This algorithm permits the identification of fragments of protein that can be recombined safely without the disruption of the three-dimensional integrity of a protein [92]. SCHEMA calculates the interactions between a protein’s amino acids and evaluates which disruptions will lead to a stable recombinant protein. Ojala et al. [93] applied SCHEMA and RASPP [94] algorithms to design the library of chimeric AAV with the increased transduction of adult neural stem cells (NSCs) in the subventricular zone (SVZ). Eight CAP gene recombination sites and six parental strains led to a library with 1.6 million possible variants. To select chimeric capsids with the best NSC transduction properties, AAV libraries were administered to GFAP-Cre 73.12 mice via intracerebroventricular (i.c.v.) injection. After three rounds of selection, the variant SCH9 was evaluated as one of the best: it had increased tropism to NSCs in comparison to AAV9, bound to both heparan sulphate and galactose receptors on target cells. SCH9 also covered about 58% of the final amount of the selected library. Hence, SCHEMA is a prospective tool for AAV recombinant variant production when crystal structures of parental capsids are available [93]. SCHEMA also facilitated blood–brain barrier (BBB) penetrative AAV variant development. 

Rather than direct CNS injections, intravenous injection is more practical and safe; hence, the development of a BBB penetrative capsid is preferable. Deverman et al. [95] applied SCHEMA and CREATE to produce variants with the increased transduction of CNS after intravenous injection. Parental capsid was modified with 7-mer peptide insertions into the I-588 site of VP1 protein. The AAV-PHP.B variant was claimed to be one of the best in comparison to parental AAV9 and many other library members. The relative increase in transduction was 40-fold higher than AAV9; AAV-PHP.B was able to transduce the majority of astrocytes and neurons among different regions of the CNS and passed through the BBB [95,96].

CREATE indeed is a very useful tool for library selection, but it possesses one important drawback, which consists of the inequality of capsid variant diversity, as some capsid fragments tend to be represented more frequently than others in the final capsid library. Such a disproportion may arise from amplification biases, capsid assembly efficacy, genome packaging and many other factors. Taking this disadvantage into account, a modification of the CREATE (M-CREATE) tool was developed including NGS analysis, which helps to analyze tropism variants within just one capsid type and the most suitable capsid variants for certain tissue tropisms. The M-CREATE method revealed several novel AAV variants with increased CNS transduction and to analyze their 7-mer insertion amino acid distribution in the key positions [97].

Ancestral reconstruction is also a useful method for AAV capsid engineering, as it provides the opportunity to obtain unique novel capsids, sometimes even without the usual wild-type receptors. Using Bayesian Markov chain Monte Carlo and 52 AAV sequences, extant phylogenetic tree variants were generated. A prediction of amino acid residue probabilities made it possible to identify 32 variable residues which were used to obtain a novel variant library. These variants were similar in regard to the transduction of several cell lines, but transduced muscle cells more effectively in comparison to AAV1 and did not use sialic acid, HSPC or galactose as receptors for binding [98]. Zinn et al. [99] obtained the Anc80 variant via ancestral reconstruction of 75 AAV sequences. Based on evaluated posterior probabilities for amino acids in certain positions, a library of Anc80 variants was obtained, and its properties were tested on adult mice and non-human primates. Anc80 transduced the muscle, liver and retina more effectively in comparison to wild-type AAV2 and AAV8. Anc80 also showed good results in murine models of Usher syndrome, successfully delivering the USH1C gene to improve symptoms, such as hearing loss and balance behavior [100]. Nonetheless, taking into account the fact that ancestral capsids may lack natural AAV strains, crucial features necessary for clinical application, there are still many unanswered questions.

Directed evolution is a robust approach that should be applied in the case of the absence of structural data to obtain the whole spectrum of possible capsids. Although it is time consuming, combining the evolutional approach with the rational one is useful to reduce the library’s size and the total amount of work. A Cre-dependent selection is also a good choice for the fast and precise selection of desired capsids, but there may be difficulties with the selection of any type of desired cell culture as it requires tissue-specific promoters, the construction of which is far from being a trivial task.

### 3.3. Chemical Conjugation

The third strategy involves the chemical modification of AAV with a purpose to covalently bind specific molecules, such as fluorophores, antibodies or targeting peptides (Figure 4). This approach is extremely useful in cases when the insertion size is limited due to the possibility of incorrect capsid assembly and the inability to assemble.

For example, an avidin–biotin complex was utilized to conjugate AAV2 capsid, with a targeting protein. Epidermal growth factor (EGF) and fibroblast growth factor 1α (FGF1α) were conjugated with a core streptavidin for targeting specific receptors and conjugation with the capsid. The biotinylation of AAV2 capsid was performed with an NHS–biotin complex (N-Hydroxy-succinimide ester–water-soluble biotin). The subsequent incubation of the EGF– or FGF1α–streptavidin complex and biotinylated AAV2 resulted in the conjugation of targeting peptides with capsids. Tropism-modified capsid showed increased transduction in EGFR-positive SKOV3.ip1 cells, and FGF1α-containing AAV2 successfully transduced M07e cells, which are typically resistant to AAV2 transduction [101].

In another case, an aldehyde tag was introduced into the aforementioned I587 insertion site. Inserted 13-amino acid peptide LCTPSRAALLTGR contained cysteine, which contains the thiol group. Subsequent incubation with formylglycine-generating enzyme (FGE) led to the conversion of cysteine to aldehyde-containing formylglycine. Hence, AAV capsid contained aldehyde groups on the surface, which can be used for the covalent binding of hydrazide- and aminooxy-containing antibodies, proteins, nanoparticles, etc. The chemical binding of the RGD motif to AAV2 capsid resulted in a ~3-fold robust increase in HeLa cells, and binding with anti-HLA mouse antibody increased 293T cell transduction from 15.7% to 35.7% in comparison with wild-type AAV2 [102]. 

An intriguing technique of unnatural amino acid (UAA) insertion for a fine bioconjugation was exploited. Owing to an abundance of lysine and arginine, which are the main targets for bioconjugation, the nonselective alteration of capsid amino acids may lead to the modification of residues that are crucial for viral entry. Hence, the application of UAA may help to avoid problems with both the virus entry and assembly. Non-canonical amino acid AzK was incorporated into AAV capsid instead of five tropism-related residues: G453, T454, N587, D327 and R588. After the incorporation and subsequent chemical modification of AzK, the RGD motif was conjugated with capsids and exposed to cell cultures SK-OV-3 and HEK293. The R588 variant showed an increased transduction of SK-OV-3 cells [103]. 

DNA–protein interactions may also be used to conjugate targeting proteins with capsids. For example, HUH-tag (21kDa), which is known for its ability to bind specific single-stranded DNA sequences, was introduced into AAV capsid at the variable IV region. This DNA sequence was then covalently linked to an anti-GFP antibody with a copper-free click chemistry technique. The resulting AAV–HUH–antibody capsids were then incubated with a surface GFP-expressing HEK293 culture. As a result, AAV–HUH–antibody complex transduced HEK293 cells more effectively in comparison to the non-complex AAV. Another test on neuronal cells showed that AAV–HUV–L1CAM effectively transduced neuronal cells, but poorly transduced glial cells, as L1CAM is not expressed by these cells [104].

## 4. Discussion

The present review shows the most useful and effective approaches for AAV tropism modification: rational design, directed evolution and chemical conjugation. A huge amount of work devoted to AAV tropism modification succeeded in the construction of novel AAV capsids for different purposes: the transduction of muscle tissue, CNS, liver, retina, etc. Over the last three decades, great progress has been made, and modern techniques allow novel capsids to be discovered much faster than before. Studies of post-translational modifications and capsid immunogenic sites should also be taken into account. Acquiring more information about both the influence of such capsid protein modifications and immunogenic sites could facilitate the rational design of viral vectors that is to become the dominant approach in the distant future. This branch of investigations is necessary for the improvement of AAV properties, and further work will increase the amount of AAV-based approved therapies.

## 5. Expert Opinion

AAV is a prospective tool for gene therapy, but known serotypes are usually unsuitable for direct clinical applications due to their low specificity and immune response, which includes both cytotoxic lymphocytes and neutralizing antibodies. Several AAV-based drugs have been approved for clinical application, but serotypes used for gene delivery possess natural tropism to target tissues: Luxturna, which was developed for the treatment of Leber congenital amaurosis, is based on the AAV2 serotype that has natural tropism to the retina [26]; Zolgensma, which utilizes AAV9 for the treatment of spinal muscular atrophy [42]; and Glybera, which was also approved for lipoprotein lipase deficiency treatment via muscle transduction with the AAV1 serotype [12]. The development of such therapy should theoretically decrease the costs of treatment due to the relatively long expression of therapeutic plasmid. However, AAV’s drawbacks pose serious obstacles for the successful implementation of gene therapy in clinical practice, and both of them can be overcome with capsid engineering research. Comprehensive knowledge of a tissue’s proteome is also crucial, as it is sometimes problematic to define a cell-specific transmembrane protein that could be used for a targeted AAV-based transduction. Modern research has become more advanced in comparison to pioneering works through the use of bioinformatics calculations which allow the prediction of optimal recombination sites, analyze capsid structures and describe capsid–receptor interactions. However, while structural analysis helps to reveal an insertion site on the surface of a capsid, experimental tests sometimes show that insertion does not increase the transduction level. Thus, today, the rational approach is limited to only well-studied sites, not always allowing the insertion of a peptide of interest due to the restriction of the peptide’s length. Rational design can be extremely useful, as it helps to decrease the amount of work and to narrow the search area for proper AAV variants, especially when being combined with other approaches.

Directed evolution creates the opportunity to generate diverse mutants with unique properties, even in the case of the absence of biological data, and modern techniques allow the selection of suitable variants from a huge library in vivo. Since pioneering work in this area, directed evolution has become more effective and useful, especially when combined with bioinformatics tools. However, although novel methods of directed evolution include different tools to decrease the amount of work, it is still time consuming and requires accurate calculations for the best recombination and mutation sites.

Chemical conjugation is also a suitable approach that becomes especially useful when there is a high possibility to disrupt capsid assembly or cellular traffic due to the large size of the inserted sequence. It also seems to be less expensive and time consuming in comparison to the aforementioned approaches, but still requires data on suitable target–ligand pairs with strong bonds. Nevertheless, in some cases, it is more logical to use covalent binding, especially in the presence of targeting antibodies and proper conjugation reagents. 

While many works have been devoted to direct capsid engineering, it may be useful to study the post-translational modifications of the AAV capsid to achieve better AAV clinical properties. Engineering AAV capsids to evade the immune response also warrants investigation. Depending on the serotype, 30 to 70% of the human population have neutralizing antibodies to AAV [105]. Ignoring this fact while developing AAV-based therapeutic tools may lead to failures. Modern research has gained some data on this subject and allowed an increase in the transduction efficiency of AAV2.5 and AAV-DJ variants via both retargeting and immune response evading. However, this problem is incredibly important, and there are still too many knowledge gaps to state that it has been resolved completely.

In conclusion, further investigation will definitely be focused on rational design and bioinformatics calculations, but 5–10 years is not a sufficient amount of time to significantly shift the balance towards rational design; hence, other techniques of capsid engineering are still in play and must complement one another to result in effective and safe therapy.

## Figures and Tables

**Figure 1 pharmaceutics-13-00750-f001:**
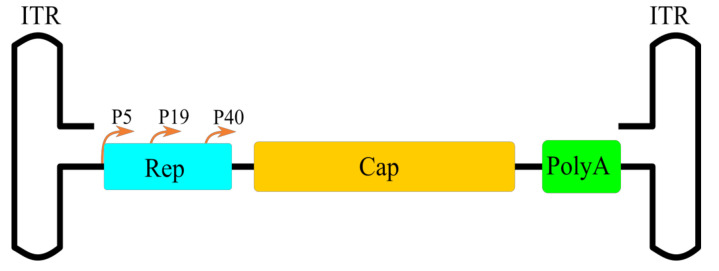
Schematic illustration of AAV genome.

**Figure 2 pharmaceutics-13-00750-f002:**
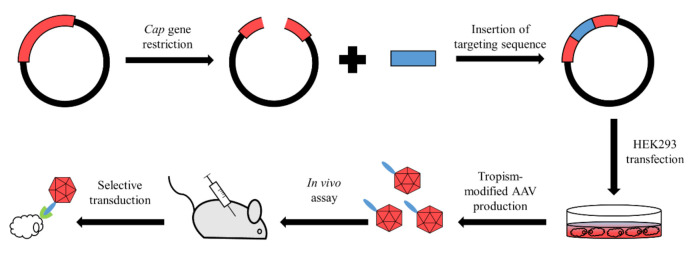
Generalized steps of AAV rational design. Red line represents Cap gene inside of plasmid. Digestion of Cap gene in the insertion site allows a targeting sequence to be plugged inside of the gene. Further steps lead to production of tropism-modified AAV which is able to transduce target cells either in or ex vivo.

**Figure 3 pharmaceutics-13-00750-f003:**
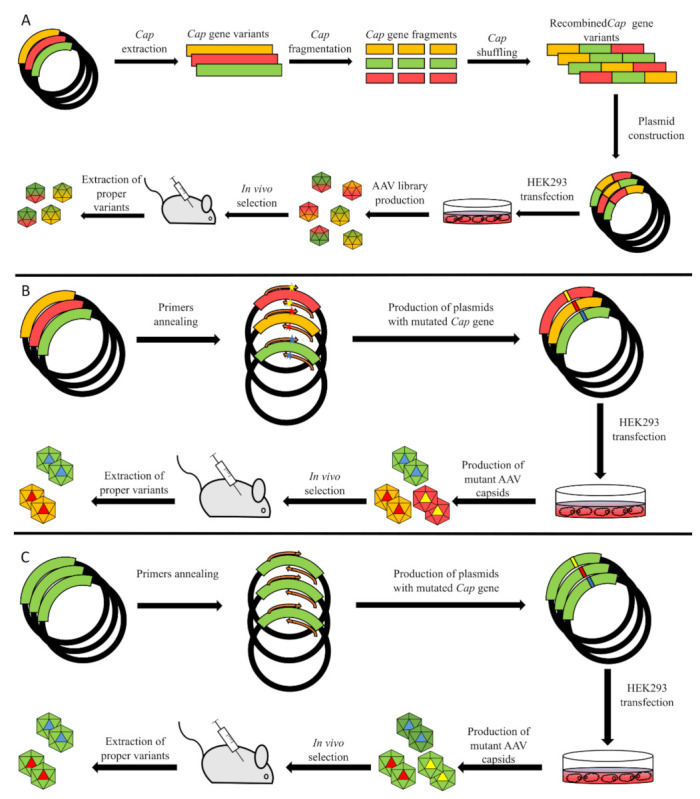
Available methods for AAV library generation. (**A**) Gene shuffling. Red, yellow and green lines represent variants of the Cap gene, which then undergoes fragmentation and shuffling of fragments. Subsequent transfection of AAV-producing cells leads to generation of novel chimeric capsids, which then undergo selection either in a model animal or in tissue culture. After selection, DNA of capsids that transduce target cells is extracted and sequenced to define which shuffled variants of the Cap gene possess the desired properties. (**B**) Site-directed mutagenesis. This method is based on the usage of primers with point mutations (illustrated as colored stars). Afterwards, PCR mutant plasmids are used for production of an AAV library which then undergoes selection. (**C**) Error-prone PCR. The main idea of error-prone PCR is to introduce random mutations into a gene of interest. This can be done with low-fidelity polymerases or with addition of chemical compounds that additionally decrease fidelity of an enzyme.

**Figure 4 pharmaceutics-13-00750-f004:**
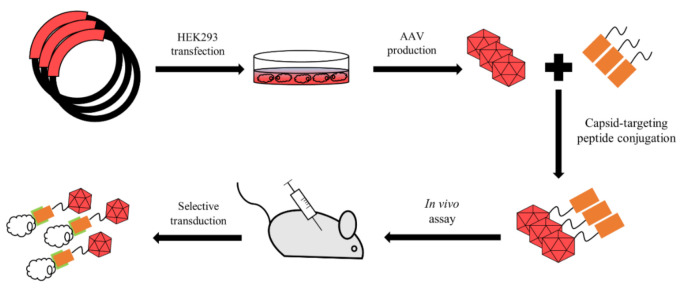
AAV bioconjugation. Producing cell line undergoes transfection with a plasmid encoding capsid of desired strains. Extracted virus is then conjugated with a targeting molecule via covalent bond. Conjugated capsid should be able to transduce target cells.

**Table 1 pharmaceutics-13-00750-t001:** Characterization of AAV natural serotypes.

Serotype	Origin	Primary Receptor	Secondary Receptor	Natural Tropism	Selected Ongoing Clinical Trials	Links
**AAV1**	Non-human primate	Sialic acid	AAV receptor (AAVR)	Muscle, CNS, heart, liver, lungs	No trials underway	[10,11,12,13]
**AAV2**	Human	Heparan sulfate proteoglycan (HSPG)	Integrin, fibroblast growth factor receptor (FGFR), hepatocyte growth factor receptor (HGFR), laminin receptor (LamR)	Heart, CNS, liver, lungs, retina	Pompe disease (NCT03533673), Parkinson’s disease (NCT01621581), hemophilia (NCT03489291)	[14,15,16,17,18,19,20,21]
**AAV3**	Non-human primate	HSPG	LamR, FGFR, HGFR, AAVR	Liver	No trials underway	[22,23]
**AAV4**	Non-human primate	Sialic acid	Unknown	Retina, lungs, kidney	No trials underway	[24,25]
**AAV5**	Human	Sialic acid	Platelet-derived growth factor receptor (PDGFR), AAVR	Retina, CNS, liver	Hemophilia (NCT03520712)	[13,16,26,27,28,29,30]
**AAV6**	Human	HSPG, sialic acid	EGFR, AAVR	Heart, liver, muscle, retina	Hemophilia (NCT03061201) mucopolysaccharidosis type I (NCT02702115)	[31,32,33]
**AAV7**	Non-human primate	Unknown	Unknown	Liver		[25]
**AAV8**	Non-human primate	Unknown	LamR, AAVR	Muscle, heart, CNS, liver	Eye disease (NCT03066258), hemophilia (NCT00979238), myopathy (NCT03199469)	[21,30,34,35,36]
**AAV9**	Human	Galactose	LamR, AAVR	Heart, CNS, liver	Muscle diseases (NCT03362502), Pompe disease (NCT02240407), Danon disease (NCT03489291)	[25,36,37,38,39,40,41,42]
**AAV10**	Non-human primate	Unknown	Unknown	Muscle, myoblast tissue	No trials underway	[43]
**AAV11**	Non-human primate	Unknown	Unknown	Muscle, myoblast tissue	No trials underway	[43]
**AAV12**	Non-human primate	Unknown	Unknown	Salivary glands, muscle	No trials underway	[44]

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
