# Peer review of "Next Step in Gene Delivery: Modern Approaches and Further Perspectives of AAV Tropism Modification"

_pharmaceutics, 2021, doi:10.3390/pharmaceutics13050750_

Round 1

Reviewer 1 Report

The authors have done a great job in summarizing the field. The authors have covered all major aspects. The discussion and expert opinion sections are informative. However, some discussions of engineering to evade immune responses might be useful. Also, table 1 could possibly use a list of current clinical trials.

Author Response

We thank the referee for the kind review and helpful comments.

However, some discussions of engineering to evade immune responses might be useful.

As it was recommended, we have added discussions concerning AAV engineering to evade immune responses into the sections “4. Discussion” and “5. Expert opinion”. Additional text in both sections is colored red.

Also, table 1 could possibly use a list of current clinical trials.

We have included the data on clinical trials of AAV-based drugs into Table 1 as requested by the reviewer. Additional text is also colored red.

Reviewer 2 Report

Topic is interesting and broad prospects with background have been discussed and presented well. English language must be improved and the review can be accepted in the current form.

Author Response

We thank the reviewer for the positive evaluation of our manuscript. Additional English editing was performed by MDPI-based service.